

# A pan-African high-resolution drought index dataset

**Jian Peng [1,2], Simon Dadson [1], Feyera Hirpa [1], Ellen Dyer [1], Thomas Lees [1], Diego G. Miralles [3], Sergio M. Vicente-Serrano [4], Chris Funk [5,6]**

1. School of Geography and the Environment, University of Oxford, OX1 3QY Oxford, UK;

2. Max Planck Institute for Meteorology, Hamburg, Germany;

3. Laboratory of Hydrology and Water Management, Ghent University, Ghent, Belgium;

4. Instituto Pirenaico de Ecología, Consejo Superior de Investigaciones Científicas (IPE-CSIC) Zaragoza, Spain;

5. U.S. Geological Survey, Earth Resources Observation and Science Center, Sioux Falls, South Dakota;

6. Santa Barbara Climate Hazards Center, University of California, USA;

Corresponding author: Jian Peng (jian.peng@ouce.ox.ac.uk)

**Abstract**

Droughts in Africa cause severe problems such as crop failure, food shortages, famine, epidemics and even mass migration. To minimize the effects of drought on water and food security over Africa, a high-resolution drought dataset is essential to establish robust drought hazard probabilities and to assess drought vulnerability considering a multi- and cross-sectorial perspective that includes crops, hydrological systems, rangeland, and environmental systems. Such assessments are essential for policy makers, their advisors, and other stakeholders to respond to the pressing humanitarian issues caused by these environmental hazards. In this study, a high spatial resolution Standardized Precipitation-Evapotranspiration Index (SPEI) drought dataset is presented to support these assessments. We compute historical SPEI data based on Climate Hazards group InfraRed Precipitation with Station data (CHIRPS) precipitation estimates and Global Land Evaporation Amsterdam Model (GLEAM) potential evaporation estimates. The high resolution SPEI dataset (SPEI-HR) presented here spans from 1981 to 2016 (36 years) with 5 km spatial resolution over the whole Africa. To facilitate the diagnosis of droughts of different durations, accumulation periods from 1 to 48 months are provided. The quality of the resulting dataset was compared with coarse-resolution SPEI based on Climatic Research Unit (CRU) Time-Series (TS) datasets, and Normalized Difference Vegetation Index (NDVI) calculated from the Global Inventory Monitoring and Modeling System (GIMMS) project, as well as with root zone soil moisture modelled by GLEAM. Agreement found between coarse resolution SPEI



from CRU TS (SPEI-CRU) and the developed SPEI-HR provides confidence in the estimation of temporal
and spatial variability of droughts in Africa with SPEI-HR. In addition, agreement of SPEI-HR versus NDVI
and root zone soil moisture – with average correlation coefficient (R) of 0.54 and 0.77, respectively – further
implies that SPEI-HR can provide valuable information to study drought-related processes and societal
impacts at sub-basin and district scales in Africa. The dataset is archived in Centre for Environmental Data
Analysis (CEDA) with link: http://dx.doi.org/10.5285/bbdfd09a04304158b366777eba0d2aeb (Peng et al.,
2019a)
**Keywords:**
Drought, Africa, Drought index, High resolution, Precipitation, Potential evaporation, drought management,
disaster risk reduction
























## 1 Introduction

Drought is a complex phenomenon that affects natural environments and socioeconomic systems in the
world (Van Loon, 2015; Vicente-Serrano, 2007; von Hardenberg et al., 2001; Wilhite and Pulwarty, 2017).
Impacts include crop failure, food shortage, famine, epidemics and even mass migration (Ding et al., 2011;
Wilhite et al., 2007; Zhou et al., 2018). In recent years, severe events have occurred across the world, such as
the 2003 central Europe drought (García-Herrera et al., 2010), the 2010 Russian drought (Spinoni et al.,
2015), the 2011 Horn of Africa drought (Nicholson, 2014), the southeast Australian's Millennium drought
(van Dijk et al., 2013), the 2013/2014 California drought (Swain et al., 2014), the 2014 North China drought
(Wang and He, 2015) and the 2015–2017 Southern Africa drought (Baudoin et al., 2017; Muller, 2018).
Widespread negative effects of these droughts on natural and socioeconomic systems have been reported
afterwards (Arpe et al., 2012; Griffin and Anchukaitis, 2014; Mann and Gleick, 2015; Wegren, 2011). Thus,
there is a clear need to improve our knowledge about the spatial and temporal variability of drought, which
provides a basis for quantifying drought impacts and the exposure of society, the economy and the
environment over different areas and time-scales (AghaKouchak et al., 2015).
Generally, drought is defined as a temporal anomaly characterized by a deficit of water compared with long-
term conditions (Mishra and Singh, 2010; Van Loon, 2015). Droughts can typically be grouped into five
types: meteorological (precipitation deficiency), agricultural (soil moisture deficiency), hydrological (runoff,
groundwater deficiency), socioeconomic (social response to water supply and demand) and environmental or
ecologic (AghaKouchak et al., 2015; Crausbay et al., 2017; Keyantash and Dracup, 2002). These different
drought categories involve different event characteristics in terms of timing, intensity, duration, and spatial
extent, making it very difficult to characterize droughts quantitatively (Lloyd-Hughes, 2014; Panu and
Sharma, 2002; Vicente-Serrano, 2016). For this reason numerous drought indices have been proposed for
precise applications, and reviews of the available indices have been provided by previous studies such as
Heim Jr (2002), Keyantash and Dracup (2002), and Mukherjee et al. (2018). Van Loon (2015) noted that





there is no best drought index for all types of droughts, because every index is designed for a specific
drought type, thus multiple indices are required to capture the multifaceted nature of drought. Nevertheless,
the Standardized Precipitation Index (SPI) is recommended by the World Meteorological Organization
(WMO) for drought monitoring, which is calculated based solely on long-term precipitation data over
different time spans (McKee et al., 1993). The advantages of SPI are its relative simplicity and its ability to
characterize different types of droughts given the different times of response of different usable water
sources to precipitation deficits (Kumar et al., 2016; Zhao et al., 2017). However, information on
precipitation is not enough to characterize drought; in most definitions, drought conditions also depend on
the demand of water vapor from the atmosphere. More recently, Vicente-Serrano et al. (2010) proposed an
alternative drought index for SPI, which is called Standardized Precipitation Evapotranspiration Index
(SPEI). Compared to SPI, it considers not only the precipitation supply, but also the atmospheric evaporative
demand (Beguería et al., 2010; Vicente-Serrano et al., 2012b). This makes the index more informative of the
actual drought effects over various natural systems and socioeconomic sectors (Bachmair et al., 2016;
Bachmair et al., 2018; Kumar et al., 2016; Peña-Gallardo et al., 2018a; Peña-Gallardo et al., 2018b; Sun et
al., 2018; Sun et al., 2016c; Vicente-Serrano et al., 2012b).
For the calculation of SPEI, high-quality and long-term observations of precipitation and atmospheric
evaporative demand are necessary. These observations may either come from ground-based station data or
gridded data such as satellite and reanalysis datasets. For example, the SPEIbase (Beguería et al., 2010) and
the Global Precipitation Climatology Centre Drought Index (GPCC-DI) (Ziese et al., 2014) both provide
SPEI datasets at global scale. The SPEIbase provides gridded SPEI with a 50-km spatial resolution, and is
calculated from Climatic Research Unit (CRU) Time-Series (TS) datasets, which are produced based on
measurements from more than 4000 ground-based weather stations over the world (Harris et al., 2014). The
SPEI dataset provided by GPCC-DI has spatial resolution of 1°, and was generated from GPCC precipitation
(Becker et al., 2013; Schneider et al., 2016) and National Oceanic and Atmospheric Administration
(NOAA)'s Climate Prediction Center (CPC) temperature dataset (Fan and Van den Dool, 2008). Both of



these datasets have been applied for various drought related studies at global and regional scales (e.g., Chen
et al., 2013; Deo et al., 2017; Isbell et al., 2015; Sun et al., 2016a; Vicente-Serrano et al., 2016; Vicente-
Serrano et al., 2013). However, these global SPEI data sets' spatial resolution are too coarse to be applied at
district or sub-basin scales (Vicente-Serrano et al., 2017). A sub-basin scale quantification of drought
conditions is particularly crucial in regions such as Africa, in which geospatial data and drought indices can
be essential to manage existing drought-related risks (Vicente-Serrano et al., 2012a) and where in-situ
measurements are scarce (Anghileri et al.; Masih et al., 2014; Trambauer et al., 2013). Over last century,
Africa has been severely influenced by intense drought events, which has led to food shortages and famine
in many countries (Anderson et al., 2012; Awange et al., 2016; Funk et al., 2018; Sheffield et al., 2014;
Yuan et al., 2013). Therefore, the availability of a high-resolution drought index dataset may contribute to an
improved characterization of drought risk and vulnerability, and minimize its impact on water and food
security by supporting policy makers, water managers and stakeholders. Conveniently, with the
advancement of satellite technology, the estimation of precipitation and evaporation from remote sensing
datasets is becoming more accurate (Fisher et al., 2017). In particular, the long-term Climate Hazards group
InfraRed Precipitation with Station data (CHIRPS) (Funk et al., 2015a) precipitation and Global Land
Evaporation Amsterdam Model (GLEAM) (Miralles et al., 2011) evaporation datasets provide high-quality
datasets for near-real time drought monitoring. Here, we use CHIRPS and GLEAM datasets to develop a
pan-African high spatial resolution (5-km) SPEI dataset, which may be useful to inform drought relief
management strategies for the continent. The dataset covers the period from 1981 to 2016 and it is
comprehensively inter-compared with soil moisture, vegetation index and coarse resolution SPEI datasets.
**2 Data and Methodology**
2.1 Data
2.1.1 CHIRPS
CHIRPS is a recently-developed high-resolution, daily, pentadal, dekadal, and monthly precipitation dataset
(Funk et al., 2015a). It was produced by blending a set of satellite-only precipitation values (CHIRP) with



additional monthly and pentadal station observations. The CHIRP is based on infrared cold cloud duration
(CCD) estimates calibrated with the Tropical Rainfall Measuring Mission Multi-satellite Precipitation
Analysis version 7 (TMPA 3B42 v7) and the Climate Hazards group Precipitation climatology (CHPclim)
The CHP$_{clim}$ (Funk et al., 2015a; Funk et al., 2015e) is based on station data from the Food and Agriculture
Organization (FAO) and the Global Historical Climate Network (GHCN). Compared with other global
precipitation datasets such as Multi-Source Weighted-Ensemble Precipitation (MSWEP) (Beck et al., 2017)
and Global Precipitation Climatology Project (GPCP) (Adler et al., 2003), CHIRPS has several advantages:
a long period of record, high spatial resolution (5-km), low spatial biases and low temporal latency. It has
been widely validated and applied in various applications (e.g., Duan et al., 2016; Maidment et al., 2015;
Rivera et al., 2018; Shukla et al., 2014; Zambrano-Bigiarini et al., 2017). In particular, it was recently
validated over East Africa and Mozambique and demonstrated good performance compared to other
precipitation datasets (Dinku et al., 2018; Toté et al., 2015). Furthermore, CHIRPS was specifically designed
for drought monitoring over regions with deep convective precipitation, scarce observation networks and
complex topography (Funk et al., 2014). Its high spatial resolution makes it particularly suitable for local-
scale studies, such as sub-basin drought monitoring, especially in areas with complex topography. The
detailed description of the dataset was provided by Funk et al. (2015a). In this study, daily CHIRPS
precipitation from 1981 to 2016 was used.
2.2.2 GLEAM
GLEAM is designed to estimate land surface evaporation and root-zone soil moisture from remote sensing
observations and reanalysis data (Martens et al., 2017; Miralles et al., 2011). Specifically, the Priestley-
Taylor equation is used to calculate potential evaporation within GLEAM based on near surface temperature
and net radiation, while the root zone soil moisture is obtained from a multilayer water balance driven by
precipitation observations and updated with microwave soil moisture estimates (Martens et al., 2017). The
actual evaporation is estimated by constraining potential evaporation with a multiplicative evaporative stress
factor based on root-zone soil moisture and Vegetation Optical Depth (VOD) estimates. The GLEAM





version 3a (v3a) provides global daily potential and actual evaporation, evaporative stress conditions and
root zone soil moisture from 1980 to 2018 at spatial resolution of 0.25° (Martens et al., 2017) (see
www.gleam.eu). GLEAM datasets have already been comprehensively evaluated against FLUXNET
observations and used for multiple hydro-meteorological applications (Forzieri et al., 2017; Greve et al.,
2014; Lian et al., 2018; Miralles et al., 2014; Richard et al., 2018; Vicente-Serrano et al., 2018). In particular,
two recent studies detected global drought conditions based on GLEAM potential and actual evaporation
data (Peng et al., 2019b; Vicente-Serrano et al., 2018). For this study, the GLEAM potential evaporation and
root zone soil moisture were used.
2.2.3 CRU-TS
The global gridded CRU-TS datasets provide most widely-used climate variables including precipitation,
potential evaporation, diurnal temperature range, maximum and minimum temperature, mean temperature,
frost day frequency, cloud cover and vapour pressure (Harris et al., 2014). The CRU TS datasets were
produced using angular-distance weighting (ADW) interpolation based on monthly meteorological
observations collected at ground-based stations across the world. The recently-released CRU TS version
4.0.1 covers the period 1901–2016 and provides monthly data at 50-km spatial resolution. The CRU TS
datasets have been widely used for various applications since their release (e.g., Chadwick et al., 2015;
Delworth et al., 2015; Jägermeyr et al., 2016; van der Schrier et al., 2013). The SPEIbase dataset was
generated from CRU TS datasets (Beguería et al., 2010). In this study, the CRU TS precipitation and
potential evaporation from 1981 to 2016 were used.
2.2.4 GIMMS NDVI
The Normalized Difference Vegetation Index (NDVI) can serve as a proxy of vegetation status and has been
widely applied to investigate the effects of drought on vegetation (e.g., Törnros and Menzel, 2014; Vicente-
Serrano et al., 2013; Vicente-Serrano et al., 2018). The Global Inventory Monitoring and Modeling System
(GIMMS) NDVI was generated based on Advanced Very High Resolution Radiometer (AVHRR)



observations, and has accounted for various deleterious effects such as orbital drift, calibration loss and
volcanic eruptions (Beck et al., 2011; Pinzon and Tucker, 2014). For the current study, the latest version of
GIMMS NDVI (3g.v1) was used, which covers the time period from 1981 to 2015 at biweekly temporal
resolution and 8-km spatial resolution (Pinzon and Tucker, 2014).
2.3 Methods
2.3.1 SPEI calculation
The SPEI proposed by Vicente-Serrano et al. (2010) has been used for a wide variety of agricultural,
ecological and hydro-meteorological applications (e.g., Jiang et al., 2019; Naumann et al., 2018; Schwalm et
al., 2017). It accounts for the impacts of evaporation demand on droughts and inherits the simplicity and
multi-temporal characteristics of SPI. The procedure for SPEI calculation includes the estimation of a
climatic water balance (namely the difference between precipitation and potential evaporation), the
aggregation of the climatic water balance over various time-scales (e.g., 1, 3, 6, 12, 24, or more months), and
a fitting to a certain parameter distribution. As suggested by Beguería et al. (2014) and Vicente-Serrano and
Beguería (2016), the log-logistic probability distribution is best for SPEI calculation, from which the
probability distribution of the difference between precipitation and potential evaporation can be calculated as
suggested by Vicente-Serrano et al. (2010) and Beguería et al. (2014). The negative and positive SPEI values
respectively indicate dry and wet conditions. In this study, the CHIRPS and GLEAM datasets were used for
SPEI calculation at high spatial resolution (5-km). For comparison, the SPEI at 50-km was also calculated
based on CRU TS datasets for the same 1981–2016 period. It should be noted that the SPEI over sparsely
vegetated and barren areas were masked out based on Moderate Resolution Imaging Spectroradiometer
(MODIS) land cover product (MCD12Q1) (Friedl et al., 2010), because SPEI is not reliable over these areas
(Beguería et al., 2010; Beguería et al., 2014; Zhao et al., 2017).
2.3.2 Evaluation criteria



The SPEIbase dataset (Beguería et al., 2010) was calculated with CRU TS dataset, which has been evaluated
and applied by many studies (e.g., Chen et al., 2013; Greenwood et al., 2017; Isbell et al., 2015; Sun et al.,
2016a; Um et al., 2017; Vicente-Serrano et al., 2013) The newly-generated SPEI at high spatial resolution
based on CHIRPS and GLEAM (SPEI-HR) is compared temporally and spatially with the SPEI calculated
from CRU TS datasets. In addition, the NDVI can also serve as an indicator for drought and vegetation
health, and to assess the performance of drought indices (Aadhar and Mishra, 2017; Vicente-Serrano et al.,
2013). Furthermore, root zone soil moisture is an ideal hydrological variable for agricultural (soil moisture)
drought monitoring. The recently-released root zone soil moisture (RSM) from GLEAM v3 provides a great
opportunity to evaluate whether soil moisture drought is well represented by SPEI. To facilitate direct
comparison between SPEI and NDVI as well as RSM, both NDVI and RSM are standardized by subtracting
their corresponding (1981–2016) mean and expressed the resulting anomalies as numbers of standard
deviations. This standardization has been applied by many studies to evaluate drought indices (Anderson et
al., 2011; Mu et al., 2013; Zhao et al., 2017). The correlation between SPEI and the standardized NDVI and
RSM is quantified using Pearson's correlation coefficient (R). In addition, the high resolution SPEI from
GLEAM and CHIRPS is also resampled to the same grid size of SPEI from CRU TS in order to quantify
their correlation and disentangle whether the added value of the former arises from its increased accuracy or
higher resolution. In the following part, the high (5-km) resolution SPEI is referred to SPEI-HR, while the
coarse 50-km resolution SPEI is referred to SPEI-CRU.
**3 Results and discussion**
3.1 Inter-comparison between high- and coarse-resolution SPEI
Figure 1 shows the spatial distribution of SPEI-HR and SPEI-CRU at different resolutions for an example
month (June 1995). Figure 1a,b show the 3-month SPEI and 12-month SPEI, respectively. It can be seen that
the high resolution and coarse resolution SPEI display quite similar dry and wet patterns over the whole of
Africa for both temporal scales. However, as expected, the SPEI-HR shows much more spatial detail that
reflects mesoscale geographic and climatic features, which highlights the advantages of this new dataset. The
differences in patterns between 3-month and 12-month SPEI indicate the different water deficits caused by
different aggregation time scales, which can further separate agricultural, hydrological, environmental, and
other droughts. For example, in June 1995 southern Africa showed persistent dry conditions over a
prolonged period, while western Africa only showed a short-term drought.

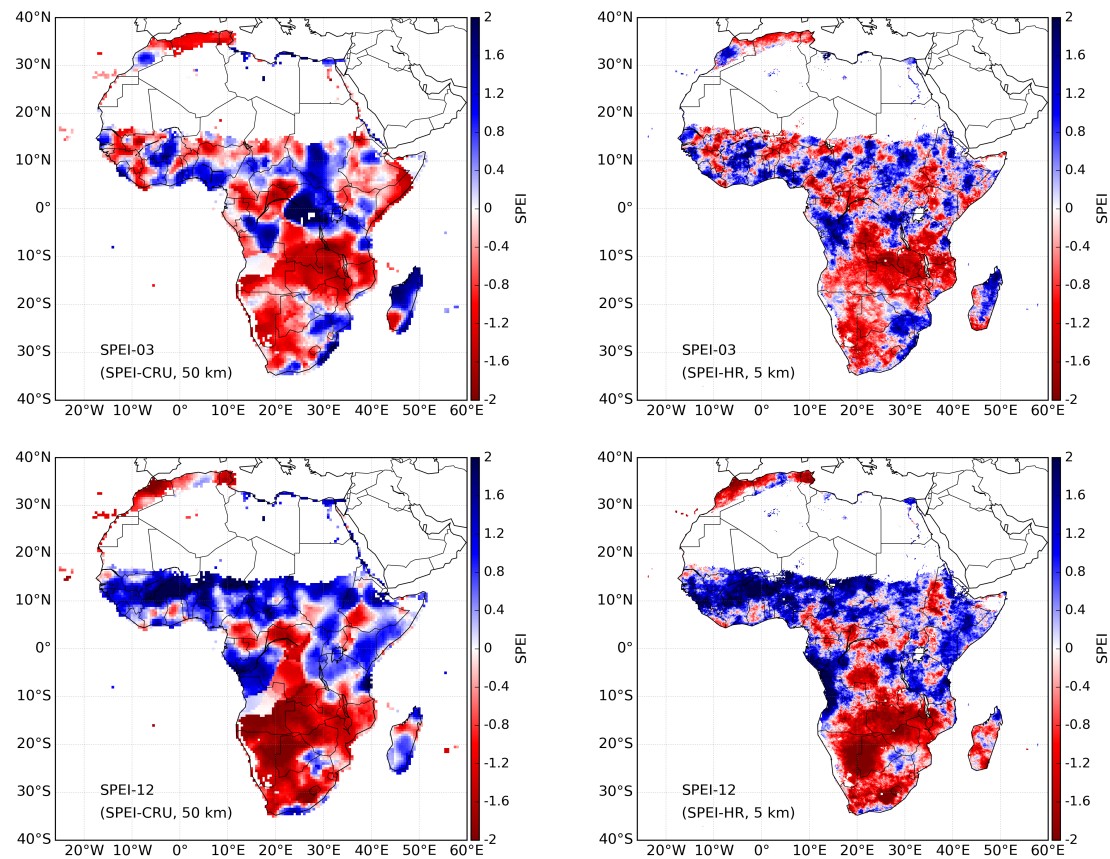

Figure 1: Spatial patterns of 3-month and 12-month SPEI at high spatial resolution (5 km) and coarse spatial resolution (50 km) in
June, 1995. The high spatial resolution SPEI (SPEI-HR) is based on CHIRPS precipitation and GLEAM potential evaporation,
while the coarse spatial resolution SPEI (SPEI-CRU) is calculated from CRU TS datasets.

In order to quantify how different is SPEI-HR from SPEI-CRU, the correlation between them is calculated
for each grid cell over the whole study period. Figure 2 shows the correlations for time-scales 1, 3, 6, 9, 12,
24, 36, and 48 months. In general, the SPEI-HR and SPEI-CRU agree well in terms of temporal variability
with high positive correlations over most of Africa for every time scale. However, relatively low correlations
appear in central Africa, and they become lower as the SPEI time-scale increases. This region has very few



station observations. It should be noted that the correlations shown here are statistically significant with p
value less than 0.05.  In addition, the average correlation between 6-month SPEI-CRU and SPEI-HR for
each month of the year is summarized in Figure 3 using box plot. In general, positive correlations, with a
median larger than 0.6 (p<0.05), are found for every month. There are no substantial differences in
correlations between different months. Figure A1 in Appendix shows additional box plots for SPEI at other
time scales.

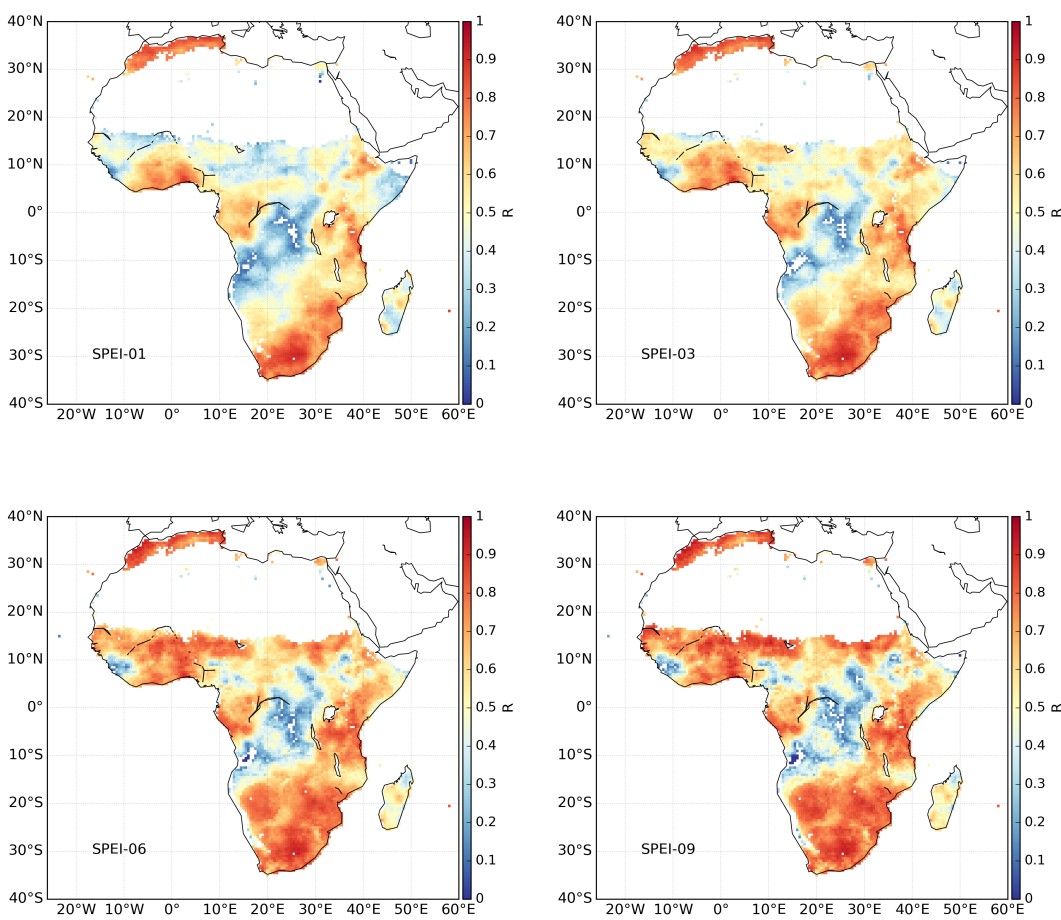

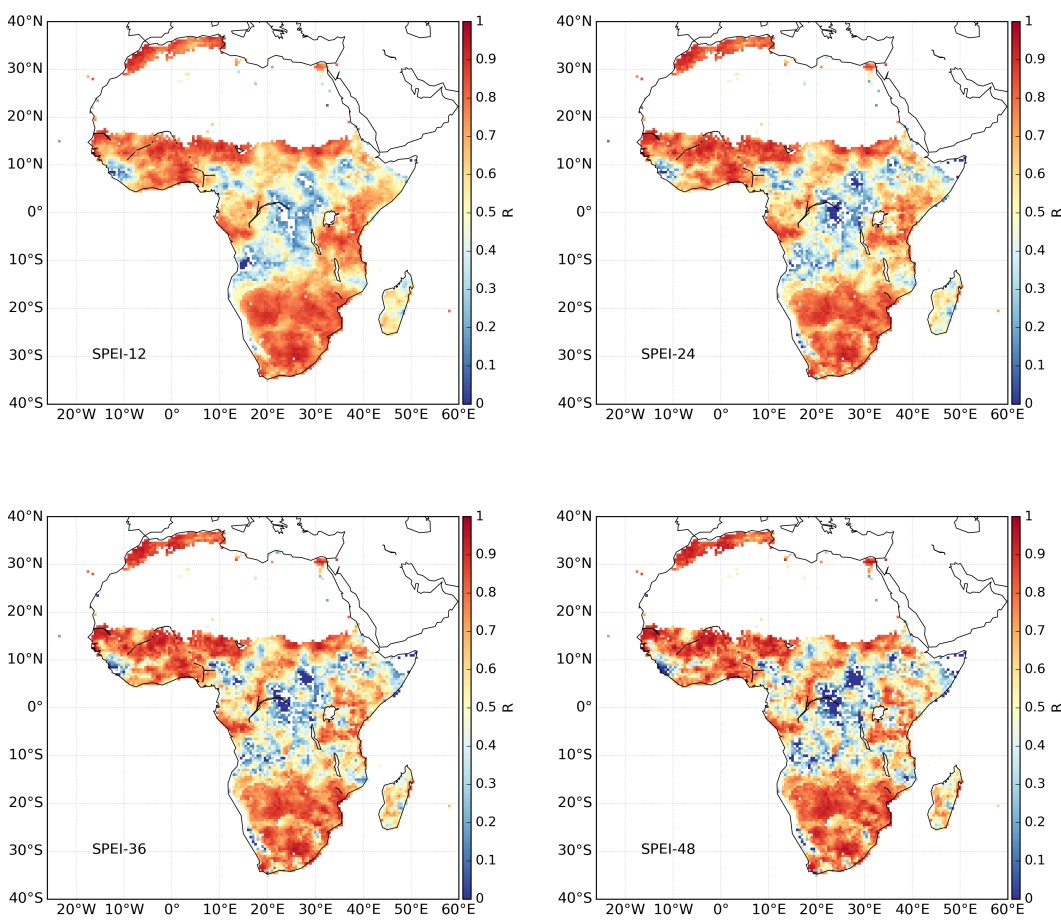

Figure 2: Correlation (p<0.05) between SPEI-HR and SPEI-CRU, with the number indicating different months.

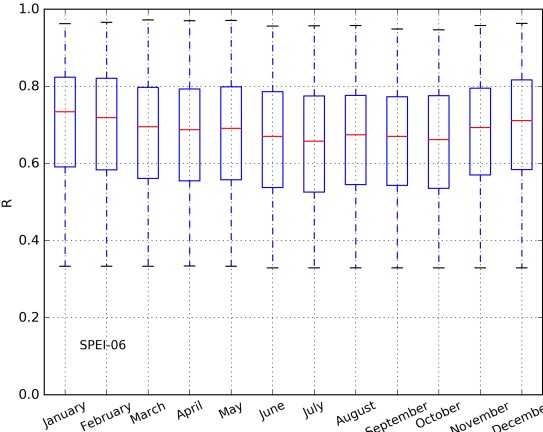

Figure 3: Box plot of the correlation (p<0.05) between SPEI-HR and SPEI-CRU for each month of the entire record. The results
here are based on 6-month SPEI and the red line in each box represents the median.

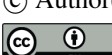




3.2 Comparison against root zone soil moisture and NDVI
To gain more insights into their significance and applicability, the SPEI datasets are compared with NDVI
and RSM. Figure 4 shows the results of the spatial and temporal comparison between 6-month SPEI and
RSM as indicated by Törnros and Menzel (2014). Figure 4a,b display the correlation (p<0.05) of SPEI-HR
and SPEI-CRU against RSM during the whole time period respectively. In general, both SPEI-HR and SPEI-
CRU show strong correlations with RSM over the whole African continent. Compared to SPEI-CRU, the
SPEI-HR shows higher correlations, particularly over central Africa. Since Section 3.1 shows that relatively
large discrepancy between SPEI-CRU and SPEI-HR exists over central Africa, the results presented here
suggest a potentially better performance of SPEI-HR compared with SPEI-CRU in this region.
The time series of SPEI and RSM, averaged over the entire study area, are shown in Figure 4c, together with
the corresponding correlations. It can be seen that both SPEI-HR and SPEI-CRU agree well with each other
and with the RSM dynamics. Consistent with the results from the spatial correlation analysis, the SPEI-HR
and SPEI-CRU show similar results when compared with RSM (R = 0.77 for SPEI-HR, R = 0.72 for SPEI-
CRU). Furthermore, the scatterplots between 6-month SPEI and RSM for the entire data record are shown in
Appendix Figure A2, where positive and significant correlations with RSM are found for both SPEI-HR (R =
0.51) and SPEI-CRU (R = 0.42). To explore the correlation between RSM and different time scales of SPEI,
Table 1 summarizes the correlation value calculated in the same way as Figure 4c. It can be seen that the
highest correlations against RSM are found at 3- and 6-month time scales. It should be noted that satellite
data-driven estimates of root zone soil moisture is more suitable for evaluating SPEI compared to satellite-
based top-layer soil moisture or reanalysis soil moisture data (Mo et al., 2011; Xu et al., 2018).



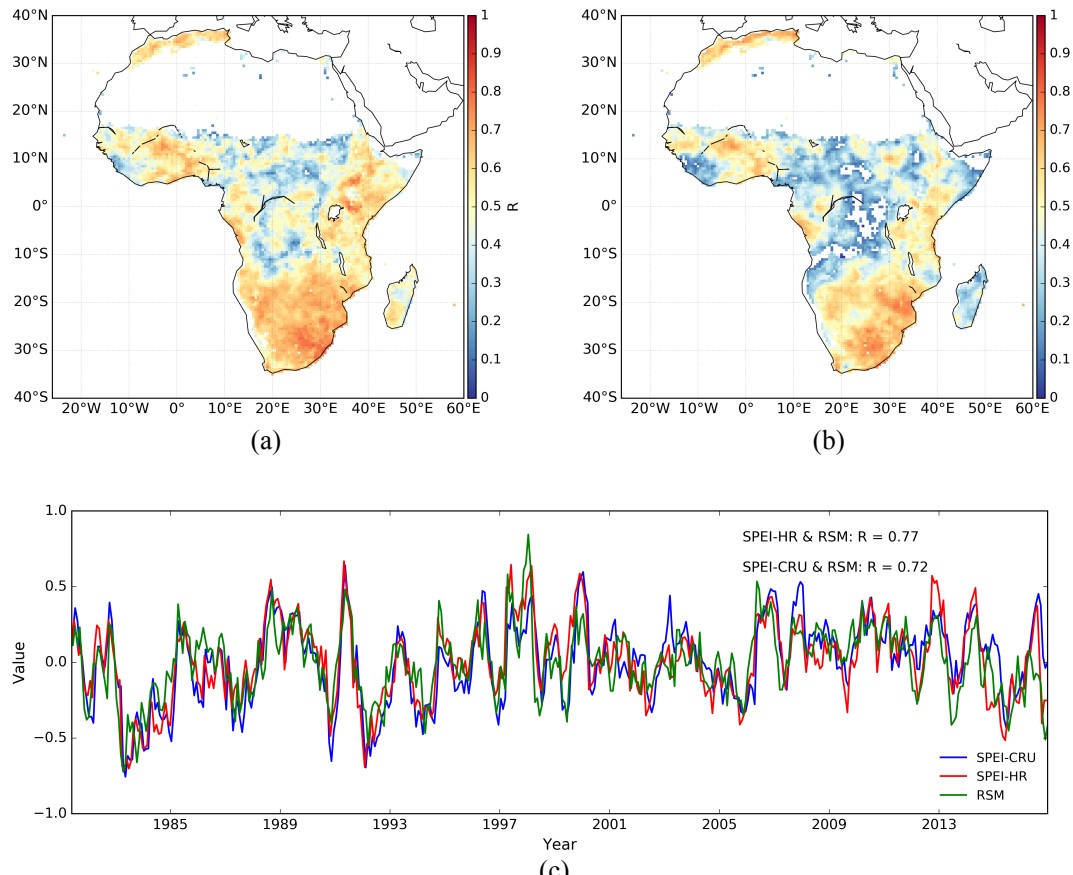

Figure 4: Spatial maps of correlation between SPEI and root zone soil moisture (RSM) for 6-month SPEI: (a) SPEI-HR and (b)
SPEI-CRU. The time series of Africa area-mean RSM and SPEI are shown in (c), where R refers to the correlation coefficient.
The correlations shown here are all significant at the 95% confidence level.
Table 1: The correlation (p<0.05) between area-mean RSM and SPEI at different time scales.

|  | SPEI-01 | SPEI-03 | SPEI-06 | SPEI-09 | SPEI-12 | SPEI-24 | SPEI-36 | SPEI-48 |
|---|---|---|---|---|---|---|---|---|
| R (SPEI-CRU) | 0.52 | 0.74 | 0.72 | 0.64 | 0.56 | 0.41 | 0.26 | 0.16 |
| R (SPEI-HR) | 0.49 | 0.76 | 0.77 | 0.69 | 0.62 | 0.44 | 0.29 | 0.18 |


Similar to the above analysis between SPEI and RSM, the comparison of results between SPEI and NDVI
are shown in Figure 5. First, Figures 5a,b present the spatial distribution of the correlations (p<0.05) between
SPEI-HR and NDVI and between SPEI-CRU and NDVI, respectively. While correlations are overall lower
than for RSM, it can be seen that both SPEI datasets are positively correlated with NDVI over most of the
continent. It is also clear that SPEI-HR shows higher correlations. The time series comparison between the



area-mean SPEI and NDVI is shown in Figure 5c. Both SPEI-HR and SPEI-CRU show agreement with
NDVI, with R=0.54 and R=0.47, respectively. In addition, the comparison between 6-month SPEI and
NDVI for the entire data record was also calculated, with R=0.24 for SPEI-HR and R=0.21 for SPEI-CRU
significant at 95% confidence level (Figure A3). While these correlations are admittedly low, overall results
suggest that the SPEI has a positive relation with NDVI, which is also reported by previous studies (e.g.,
Törnros and Menzel, 2014; Vicente-Serrano et al., 2018). The lower correlations against NDVI than against
RSM are likely due to complex physiological processes associated to vegetation, and the fact that ecosystem
state is driven by multiple variables other than water availability (Nemani et al., 2003). Furthermore, there
are also clearly documented lags between precipitation and NDVI, with NDVI time series typically peaking
one or even two months after the period of maximum rainfall (Funk and Brown, 2006). Finally, Table 2
summarizes the correlation between SPEI and NDVI at different time scales. Compared with the results
presented in Table 1 for RSM, the correlation with NDVI shown in Table 2 is also generally lower, and the
highest correlations appear between 9- and 24-month SPEI (R>0.5).

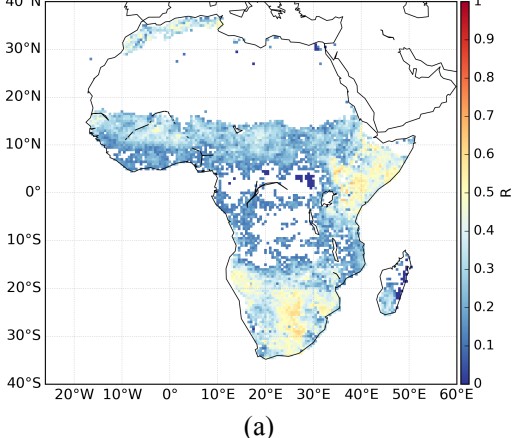

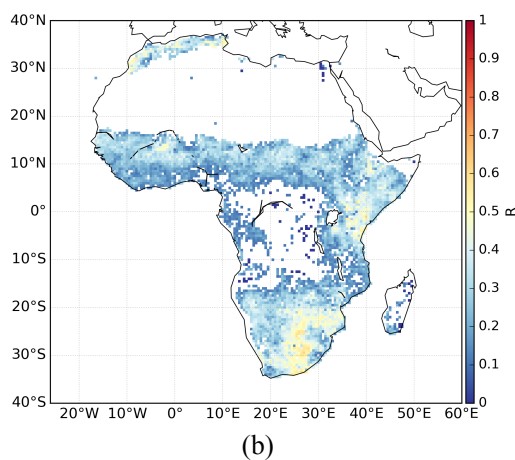

(a)                                                          (b)

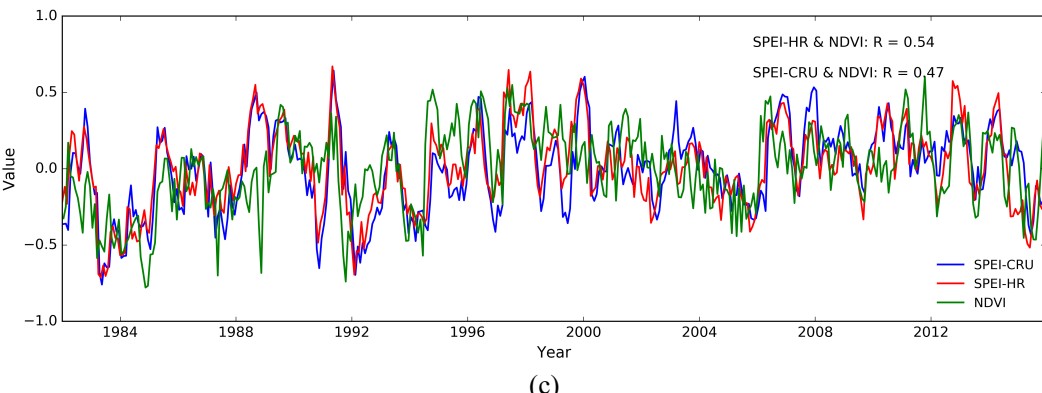

(c)

Figure 5: Spatial maps of the correlation between SPEI and NDVI for 6-month SPEI: (a) SPEI-HR and (b) SPEI-CRU. The time series of area-mean NDVI and SPEI are shown in (c), where R refers to the correlation coefficient. The correlations shown here are all significant at the 95% confidence level.

Table 2: The correlation (p<0.05) between area-mean NDVI and SPEI at different time scales.

|            | SPEI-01 | SPEI-03 | SPEI-06 | SPEI-09 | SPEI-12 | SPEI-24 | SPEI-36 | SPEI-48 |
|------------|---------|---------|---------|---------|---------|---------|---------|---------|
| R (SPEI-CRU) | 0.23  | 0.42    | 0.47    | 0.48    | 0.47    | 0.50    | 0.34    | 0.20    |
| R (SPEI-HR)  | 0.31  | 0.51    | 0.54    | 0.56    | 0.57    | 0.57    | 0.44    | 0.29    |

Altogether, the comparisons between SPEI and RSM and between SPEI and NDVI indirectly indicate the validity of the generated SPEI datasets. Therefore, the generated high-resolution SPEI-HR from satellite products has potential to improve upon the state of the art in drought assessment over Africa.

3.3 Patterns of SPEI, RSM and NDVI during specific drought events

Most of Africa has suffered severe droughts in past decades (Blamey et al., 2018; Naumann et al., 2014). Among them, the 2011 East Africa drought (AghaKouchak, 2015; Anderson et al., 2012) and 2002 southern Africa drought (Masih et al., 2014) were extremely severe and had devastating effects on the natural and socioeconomic environment. Taking these two events as case studies, the spatial patterns of the newly-developed high-resolution 6-month SPEI-HR are analyzed, together with the variability in NDVI and RSM. Figure 6a,b show the evolution of 6-month SPEI, NDVI and RSM during the 2011 East Africa and the 2002 southern Africa drought, respectively. The 6-month periods end in the named month, with the 6-month June 2011 SPEI values based on data for January to June. In general, these three variables reflect the progressive




dry-out during the events. For example, strong, severe drought is revealed by the SPEI with values less than
-1.5, coinciding with a decline in NDVI and RSM, from June to September 2011 over East Africa; the
drought was offset in October. Similarly, dry and wet conditions variations during the 2002 southern Africa
drought were also captured by the three variables. Despite differences over space and time, results here
demonstrate that the generated SPEI-HR captures the main drought conditions that are reflected by negative
anomalies in NDVI and RSM, and can thus be used to study local drought related processes and societal
impacts in Africa.

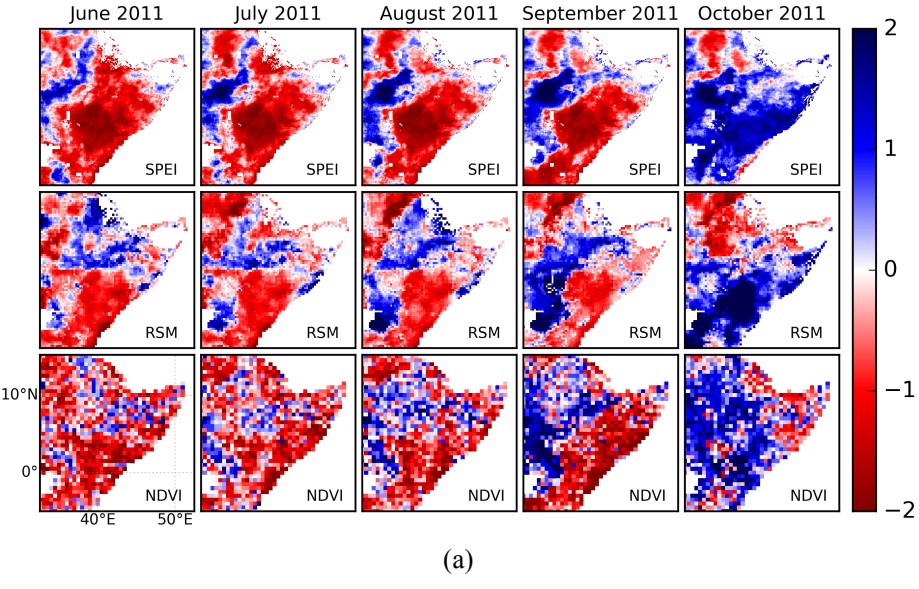

(a)

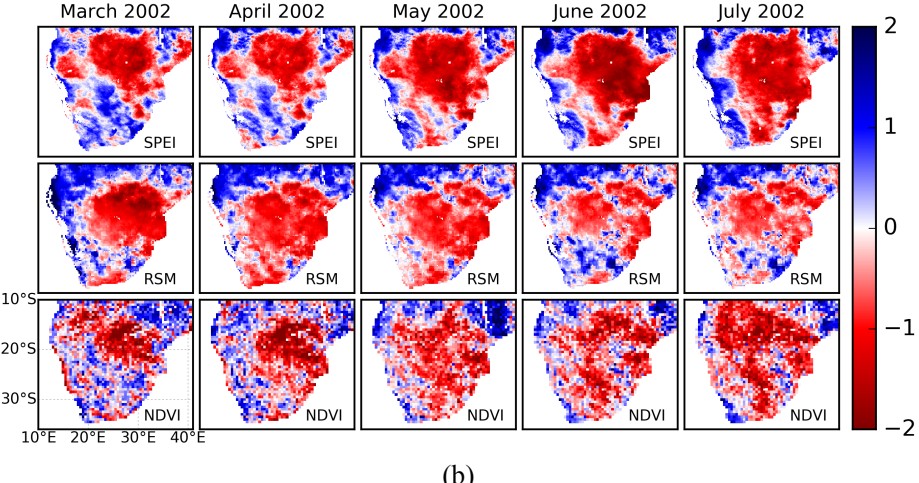

(b)




Figure 6: Evolution of the spatial patterns of 6-month SPEI-HR, NDVI and root zone soil moisture (RSM) during the 2011 East
Africa drought (a) and 2002 southern Africa drought (b), respectively.

**4. Data availability**
The high resolution SPEI dataset is publically available from the Centre for Environmental Data Analysis
(CEDA) with link: http://dx.doi.org/10.5285/bbdfd09a04304158b366777eba0d2aeb (Peng et al., 2019a). It
covers the whole Africa at monthly temporal resolution and 5 km spatial resolution from 1981 to 2016, and
is provided with Geographic Lat/Lon projection and NetCDF format.

**5. Conclusion**
The study presents a newly-generated high-resolution SPEI dataset (SPEI-HR) over Africa. The dataset is
produced from satellite-based CHIRPS precipitation and GLEAM potential evaporation, and covers the
entire African continent over the time period from 1981 to 2016 with spatial resolution of 5-km. The
accumulated SPEI ranging from 1 to 48 months is provided to facilitate applications from meteorological to
hydrological droughts. The SPEI-HR was compared with widely used coarse-resolution SPEI data (SPEI-
CRU) and GIMMS NDVI as well as GLEAM root zone soil moisture to investigate its capability for drought
detection. In general, the SPEI-HR has good correlation with SPEI-CRU temporally and spatially. They both
agree well with NDVI and root zone soil moisture, although SPEI-HR displays higher correlations overall.
These results indicate the validity and advantage of the newly developed high resolution SPEI-HR dataset,
and its unprecedentedly high spatial resolution offers important advantages for drought monitoring and
assessment at district and river basin level in Africa.






## 367 **Appendix**

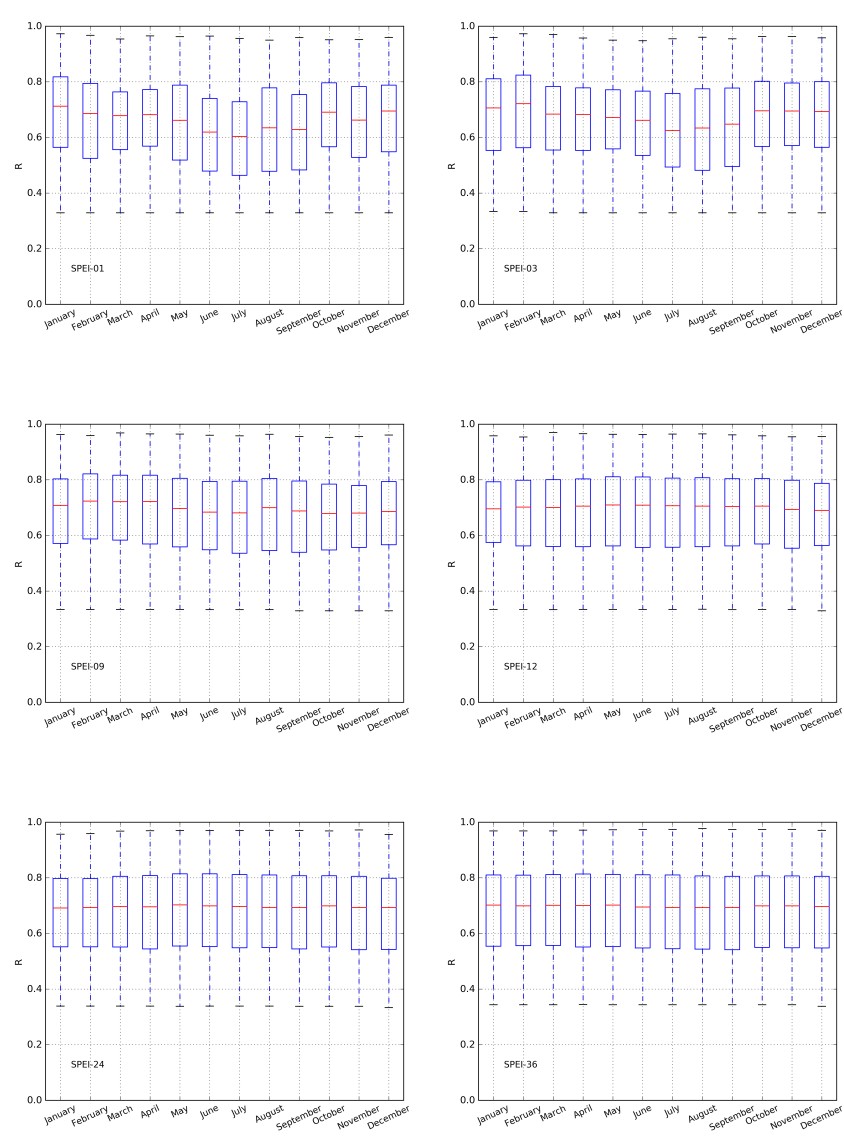

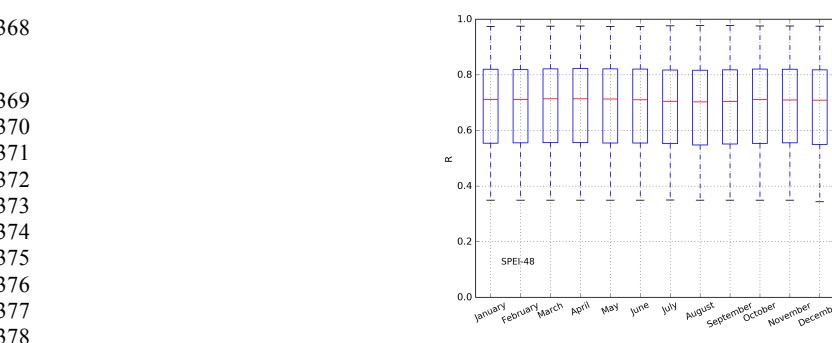

Figure A1: Box plots of the correlation (p<0.05) between SPEI-HR and SPEI-CRU for each month and entire monthly record.

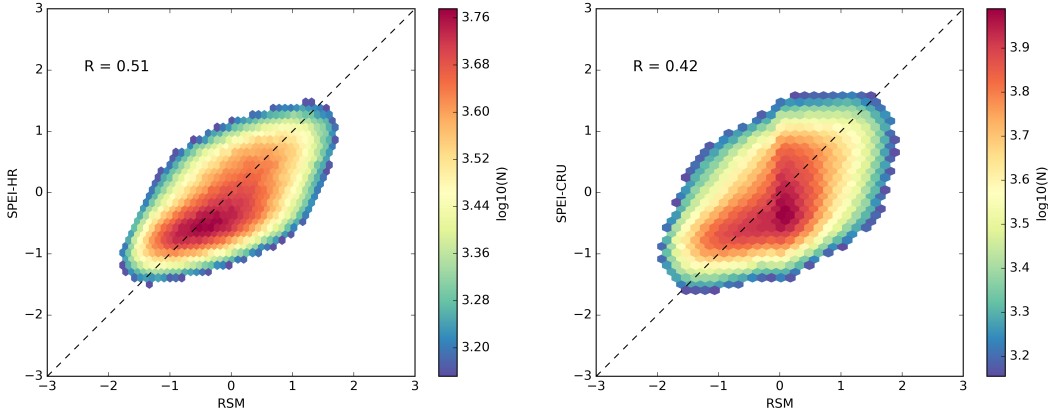

Figure A2: Scatterplots between 6-month SPEI and RSM for the entire data record. R is correlation coefficient with p<0.05, and the colors denote the occurrence frequency of values.

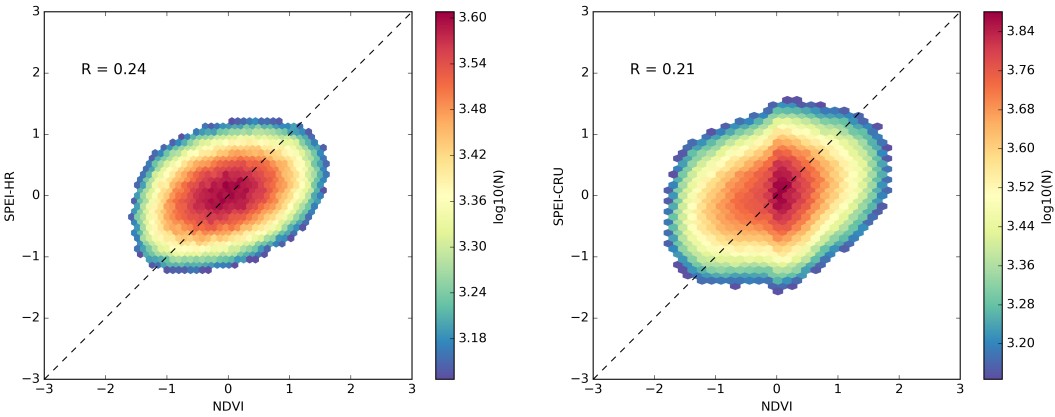



Figure A3: Scatterplots between 6-month SPEI and NDVI for the entire data record. R is correlation coefficient with p<0.05, and
the colors denote the occurrence frequency of values.

**Acknowledgments**
This work is supported by the UK Space Agency's International Partnership Programme (417000001429).
D.G.M. acknowledges funding from the European Research Council (ERC) under grant agreement 715254
(DRY–2–DRY). SD is also funded by the Natural Environment Research Council (NE/M020339/1). CF is
supported by the U.S. Geological Survey's Drivers of Drought program and NASA Harvest Program grant
Z60592017.





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
