# Peer review of "A pan-African high-resolution drought index dataset"

_Earth System Science Data, 2019_

## Referee Comment (RC1) · Anonymous Referee #1 · 22 Oct 2019

Earth Science Data

Manuscript doi: https://doi.org/10.5194/essd-2019-138

Title: A pan-African high-resolution drought index dataset

Dear editor

Thank you very much indeed for inviting me to review this paper. Having access to high-resolution drought dataset, especially in data-scarce region, is important for drought monitoring and management at watershed/ districts levels. I can be wetness that the paper "A pan-African high-resolution drought index dataset" could produce a valid significance for the African continent particularly in the drought vulnerable areas. This dataset is timely, and the paper is fully readable and has a good basis. When authors

address the following comments and suggestions, I recommend acceptance.

Comments Line 35; I couldn't get the access to the dataset. Line 38-39; delete the keywords written in the title (i.e., high-resolution, drought index) Line 78-79; insert "and/or" between "runoff, groundwater deficiency" Line 80; references should be ordered in terms of publication year and authors alphabet. And do the same for the rest in the manuscript Line 90; curiosity on using words/phrases "no best drought index", as multiscalar and multivariate drought indices are better than the single ones Line 93; change 'not enough' by 'inadequate' Line 113, curiosity on using words/phrases "too course". Line 121, Explain how the SPEI-HR dataset will be usefully to minimize the impact of water and food security and support to policymakers and the social sectors. Line 127, How can we sure that SPEI-HR can provide near-real time drought monitoring? Line 128; I have no problem with the name but I wonder why authors used Pan-Africa to represent the African continent. Does it actually represent the whole continent? Line 129; and any plan to provide data continuously in the future. Line 147; I am interested to know if your or any other studies are undertaken in Africa, using CHIRPS for drought assessment. Better if you explain why you chose this dataset for Africa. This is helpful if you refer to studies done in Africa. And the same for the potential evaporation Line 168, 179 and 188; explain why you have chosen these datasets in the context of Africa. Line 200-201, make sure 'The negative and positive SPEI values 201 respectively indicate dry and wet conditions' is correct. Line 204-205; how did you mask out and how did you manage it in your dataset Line 210, insert 'full stop (.)' after 'Vicente-Serrano et al., 2013)' Line 296, why the correlations have become low, any possible reasons Line 313, What value does the y-axis represent in figure 4 and 5 Finally, it will be very helpful if you include discussions on how the SPEI-HR is correlated with each of the drought types (meteorological, agricultural and hydrological). This can be useful to plan for short and long-term drought events mitigation based on the datasets provided.

---

## Short Comment (SC1) · 25 Oct 2019

I am eager to use this dataset. I believe that this dataset will be very valuable and helpful in Africa where data is limited. I like how the paper is written, it is very informative.

I have the following two minor comments that needs to be considered

Authors used high resolution datasets to develop the dataset. However, other criterias should have been considered. For example, supportive evidence should be provided as to whether CHIRPS is recommended for Africa or not. And the same for the other datasets used to develop this dataset. I also recommend to include the following recent researches on Africa and global to further enrich the quality of the paper. https://journals.ametsoc.org/doi/full/10.1175/BAMS-

D-12-00124.1 https://journals.ametsoc.org/doi/full/10.1175/BAMS-
D-16-0287.1 https://www.nature.com/articles/s41586-019-1149-8
https://www.sciencedirect.com/science/article/abs/pii/S0012825218303519
https://www.nature.com/articles/ngeo2646 https://onlinelibrary.wiley.com/doi/abs/10.1002/wcc.81
https://journals.ametsoc.org/doi/full/10.1175/BAMS-D-11-00176.1
https://journals.ametsoc.org/doi/full/10.1175/BAMS-D-11-00212.1
* * *

---

## Referee Comment (RC2) · Anonymous Referee #2 · 1 Nov 2019

Comments on the manuscript entitled "A pan-African high-resolution drought index dataset"

Drought is recurring and posing a certain threat to water resource and food security around the globe. Accurate and timely monitoring of droughts is essential for many applications to mitigate the potential impacts. The study aimed to generate a new high-resolution drought monitoring dataset with satellite observations, which provides a timely contribution to the scientific community. I think the produced product has a great potential to benefit drought study in Africa. To the best of my knowledge, high resolution drought dataset is not existing in the community. The widely used SPI/SPEI indices are normally based on interpolated ground measurements and have spatial resolution of 0.5 degree (∼50 km). The use of satellite products is a novel way, and

should be highly encouraged. Although 5 km is still quite coarse for agriculture applications, it might be useful for other applications e.g., regional hydrological/meteorological drought monitoring. Based on my review, I think the presented dataset adds great values for drought related applications in Africa. The manuscript is well written. The newly generated product is clearly described. I have a few fairly minor comments/suggestions below for the authors to consider for further improving the manuscript.

1. Unlike other hydrological disasters such as flood, drought is very hard to define. To this regard, there are no agreements on its definition and hundreds of drought indices have been proposed in last decades. Why do the authors choose SPEI? Why not using PDSI or others widely recognized and used index? For practical applications, how should end-user use your dataset to monitor drought? The information is missing in the manuscript, and I advise the authors to elaborate on this aspect.

2. Drought is a global disaster and deserves research at global scale. As far as I know, the satellite products used in your dataset like CHIRPS, GLEAM cover nearly entire globe (e.g. 50 dgree N-S). Why do you only focus on Africa? Why not extending to the global scale?

3. Regarding evaluation of your dataset, indirect comparison is definitely informative. Direct evaluation against ground-based measurements is essential. This part is missing in the current manuscript.

---

## Author Response (AR1)

**Response to Anonymous Referee #1**

Thank you very much indeed for inviting me to review this paper. Having access to highresolution drought dataset, especially in data-scarce region, is important for drought monitoring and management at watershed/ districts levels. I can be wetness that the paper "A pan-African high-resolution drought index dataset" could produce a valid significance for the African continent particularly in the drought vulnerable areas. This dataset is timely, and the paper is fully readable and has a good basis. When authors address the following comments and suggestions, I recommend acceptance.

Response: Many thanks indeed for your positive evaluation and constructive comments. We have revised the manuscript carefully according to your comments and suggestions. In the following, we provide an item-by-item response to your comments. Your comments are written in italic black color; our responses are shown in upright font blue color.

**Comments**

*Line 35; I couldn't get the access to the dataset.*

Response: Thanks. We have contacted CEDA team to solve the problem. The data are available now from the link.

Line 38-39; delete the key- words written in the title (i.e., high-resolution, drought index)

**Response: Done.**

*Line 78-79; insert "and/or" between "runoff, groundwater deficiency"*

**Response: Done.**

*Line 80; references should be ordered in terms of publication year and authors alphabet. And do the same for the rest in the manuscript*

**Response: Thanks, changed.**

*Line 90; curiosity on using words/phrases "no best drought index", as multiscalar and multivariate drought indices are better than the single ones*

Response: Thanks for your comment. The phrase here is reported by Van Loon (2015), which intends to note that there is no single index which is the best index and suitable for all kinds of drought events (meteorological, agricultural, hydrological, socioeconomic and environmental).

Line 93; change 'not enough' by 'inadequate'

Response: Done.

Line 113, curiosity on using words/phrases "too course".

Response: The term 'coarse' here refers to existing global products with spatial resolution of 50 km and 100 km. These datasets are not possible to provide detailed drought information at km scale that is required in district or sub-basin scale applications.

*Line 121, Explain how the SPEI-HR dataset will be usefully to minimize the impact of water and food security and support to policymakers and the social sectors.*

Response: Thanks for the comment. The important feature of SPEI-HR is its high spatial resolution compared to other coarse resolution datasets. The SPEI-HR dataset can be used to provide quantified drought conditions at sub-basin scales, which are essential for managing drought-related risks. One application of SPEI-HR for minimizing the drought impact on food security is our UK Space Agency's International Partnership Programme (417000001429). We have developed a framework to predict crop yield which can be used to infer the influence of droughts on agriculture and economics in general and specifically in Ethiopia.

Line 127, How can we sure that SPEI-HR can provide near-real time drought monitoring?

Response: The CHIRPS dataset is available from 1981 to near-real time, while GLEAM will be delivered in higher resolution and in near-real time. The idea here is to update SPEI-HR based on GHIRPS and GLEAM on a regular basis to make it near-real time.

Line 128; I have no problem with the name but I wonder why authors used Pan-Africa to represent the African continent. Does it actually represent the whole continent?

Response: It is a good question. The idea of using Pan-Africa is inspired by Pan-Africanism (https://en.wikipedia.org/wiki/Pan-Africanism). There is no difference for this study using either Pan-African or African.

*Line 129; and any plan to provide data continuously in the future.*

Response: Yes, the dataset is planned to be updated when there are new CHIRPS and GLEAM datasets released.

Line 147; I am interested to know if your or any other studies are undertaken in Africa, using CHIRPS for drought assessment. Better if you explain why you chose this dataset for Africa. This is helpful if you refer to studies done in Africa. And the same for the potential evaporation

Response: Thanks for your suggestions. The motivation of using CHIRPS for Africa is because it was recently validated over East Africa and Mozambique and demonstrated good performance compared to other precipitation datasets (Toté et al., 2015; Dinku et al., 2018). Furthermore, CHIRPS was specifically designed for drought monitoring over regions with deep convective precipitation, scarce observation networks and complex topography (Funk et al., 2014). Several studies (e.g., Toté et al., 2015; Guo et al., 2017) have used CHIRPS for drought monitoring. Similarly, GLEAM evaporation products have been widely validated/evaluated over Africa (e.g., Trambauer et al., 2014, Zhan et al., 2019). In particular, two recent studies detected global

drought conditions based on GLEAM potential and actual evaporation data (Vicente-Serrano et al., 2018; Peng et al., 2019c).

Line 168, 179 and 188; explain why you have chosen these datasets in the context of Africa.

Response: All these datasets have been validated and applied by many studies. Specifically, the GLEAM root zone soil moisture is the unique long-term root zone soil moisture product that is generated based on ESA CCI surface soil moisture. And the root zone soil moisture is more relevant to drought monitoring than satellite-based surface soil moisture. The CRU-TS datasets were used because the coarse SPEIbase dataset was produced from CRU-TS datasets. And the SPEIbase dataset has been used for drought related studies in Africa. The GIMMS NDVI dataset has been selected because it has been widely applied to investigate the effects of drought on vegetation in many areas including Africa (e.g., Rojas et al., 2011; Vicente-Serrano et al., 2013; Törnros and Menzel, 2014; Vicente-Serrano et al., 2018).

*Line 200-201, make sure 'The negative and positive SPEI values 201 respectively indicate dry and wet conditions' is correct.*

Response: Yes. The SPEI negative values indicate dry conditions while positive values correspond to wet conditions.

Line 204-205; how did you mask out and how did you manage it in your dataset

Response: The MODIS land cover product was used to mask out the sparsely vegetated and barren areas in the SPEI datasets. All the datasets were preprocessed to have same projection (geographic lat/lon) and grid size using Python.

Line 210, insert 'full stop (.)' after 'Vicente-Serrano et al., 2013)'

Response: Done, thanks.

Line 296, why the correlations have become low, any possible reasons

Response: The lower correlations against NDVI than against RSM are likely due to complex physiological processes associated to vegetation, and the fact that ecosystem state is driven by multiple variables other than water availability. Similar results have been reported by Nemani et al., 2003.

Line 313, What value does the y-axis represent in figure 4 and 5

Response: As mentioned in section 2.3.2 'To facilitate direct comparison between SPEI and NDVI as well as RSM, both NDVI and RSM are standardized by subtracting their corresponding (1981–2016) mean and expressed the resulting anomalies as numbers of standard deviations.', the y-axis has no unit and represents both SPEI and standardized NDVI and RSM.

Finally, it will be very helpful if you include discussions on how the SPEI-HR is correlated with each of the drought types (meteorological, agricultural and hydrological). This can be useful to plan for short and long-term drought events mitigation based on the datasets provided.

Response: Thanks for the suggestions. SPEI is similar to SPI when representing drought types. In general, the short time scale (e.g., 1 and 3 month) SPI/SPEI is more suitable for identifying agriculture drought. When the time scale increases, the SPI/SPEI is more relevant for hydrological drought. There are many studies using different time scales of SPI/SPEI to represent different types of droughts. In the manuscript, the sentence below describes the ability of SPI/SPEI for representing different types of droughts.

"The advantages of SPI are its relative simplicity and its ability to characterize different types of droughts given the different times of response of different usable water sources to precipitation deficits (Kumar et al., 2016; Zhao et al., 2017)."

**Response to Anonymous Referee #2**

Comments on the manuscript entitled "A pan-African high-resolution drought index dataset"

Drought is recurring and posing a certain threat to water resource and food security around the globe. Accurate and timely monitoring of droughts is essential for many applications to mitigate the potential impacts. The study aimed to generate a new high-resolution drought monitoring dataset with satellite observations, which provides a timely contribution to the scientific community. I think the produced product has a great potential to benefit drought study in Africa. To the best of my knowledge, high resolution drought dataset is not existing in the community. The widely used SPI/SPEI indices are normally based on interpolated ground measurements and have spatial resolution of 0.5 degree (~50 km). The use of satellite products is a novel way, and should be highly encouraged. Although 5 km is still quite coarse for agriculture applications, it might be useful for other applications e.g., regional hydrological/meteorological drought related applications in Africa. The manuscript is well written. The newly generated product is clearly described. I have a few fairly minor comments/suggestions below for the authors to consider for further improving the manuscript.

Response: Many thanks indeed for your positive evaluation and constructive comments. We have revised the manuscript carefully according to your comments and suggestions. In the following, we provide an item-by-item response to your comments. Your comments are written in italic black color; our responses are shown in upright font blue color.

1. Unlike other hydrological disasters such as flood, drought is very hard to define. To this regard, there are no agreements on its definition and hundreds of drought indices have been proposed in last decades. Why do the authors choose SPEI? Why not using PDSI or others widely recognized and used index? For practical applications, how should end-user use your dataset to monitor drought? The information is missing in the manuscript, and I advise the authors to elaborate on this aspect.

Response: Thanks for your comments and questions. The motivation of choosing SPEI rather than other drought index is mainly due to its relative simplicity, which allows us to produce a high spatial resolution drought dataset that entirely replies on satellite-based products. In addition, SPEI has the ability to characterize different types of droughts given the different times of response of different usable water sources to precipitation deficits (Kumar et al., 2016; Zhao et al., 2017). Regarding practical applications, there is a wide range of studies that have used SPEI for different types of droughts. In addition, the SPEI negative values indicate dry conditions while positive values correspond to wet conditions. The table below has been added in the revised manuscript to show the categories of dry and wet conditions indicated by SPEI values.

Table 1. Categories of dry and wet conditions indicated by SPEI values.

| SPEI          | Category       |
|---------------|----------------|
| 2 and above   | Extremelv wet  |
| 1.5 to 1.99   | Verv wet       |
| 1.0 to 1.49   | Moderately wet |
| -0.99 to 0.99 | Near Normal    |
| -1.0 to -1.49 | Moderately dry |
| -1.5 to -1.99 | Severelv drv   |
| -2 and less   | Extremely dry  |

2. Drought is a global disaster and deserves research at global scale. As far as I know, the satellite products used in your dataset like CHIRPS, GLEAM cover nearly entire globe (e.g. 50 dgree N-S). Why do you only focus on Africa? Why not extending to the global scale?

Response: It is a good point. Theatrically, Yes, the dataset can be extended to global scale. The current study is supported by the UK Space Agency's International Partnership Programme (417000001429), which aims to focus on Africa. However, the whole framework has been established, we can produce the SPEI-HR at any regions once there is a request from potential users.

3. Regarding evaluation of your dataset, indirect comparison is definitely informative. Direct evaluation against ground-based measurements is essential. This part is missing in the current manuscript.

Response: Thanks for the suggestion. We fully agree validation with ground-based measurement is important. However, it is very challenging to implement due to the missing of ground-based measurements for both precipitation and potential evapotranspiration. As stated in the manuscript, the CHIRPS dataset has been validated in Africa with in situ measurements. However, the ground-based potential evapotranspiration measurement is not available in Africa, which hampers the calculation of SPEI using ground-based measurements. Therefore, we use indirect comparison to present the validity of generated SPEI dataset.

**Response to Gebremedhin Haile**

I am eager to use this dataset. I believe that this dataset will be very valuable and helpful in Africa where data is limited. I like how the paper is written, it is very informative.

Response: Many thanks indeed for your positive evaluation and suggested references. We have carefully revised the manuscript according to your comments. In the following, we provide an item-by-item response to your comments. Your comments are written in italic black color; our responses are shown in upright font blue color.

I have the following two minor comments that needs to be considered

Authors used high resolution datasets to develop the dataset. However, other criterias should have been considered. For example, supportive evidence should be provided as to whether CHIRPS is recommended for Africa or not. And the same for the other datasets used to develop this dataset. I also recommend to include the following recent researches on Africa and global to further enrich the quality of the paper.

https://journals.ametsoc.org/doi/full/10.1175/BAMS-D-12-00124.1 https://journals.ametsoc.org/doi/full/10.1175/BAMS-D-16-0287.1 https://www.nature.com/articles/s41586-019-1149-8 https://www.sciencedirect.com/science/article/abs/pii/S0012825218303519 https://www.nature.com/articles/ngeo2646 https://onlinelibrary.wiley.com/doi/abs/10.1002/wc https://journals.ametsoc.org/doi/full/10.1175/BAMS-D-11-00176.1 https://journals.ametsoc.org/doi/full/10.1175/BAMS-D-11-00212.1

Response: Thanks for the comments and suggestions. We fully agree. Please see our detailed responses to Referee #1 on the motivation of choosing different products. In addition, the relevant references that support the validity of CHIRPS and other datasets in Africa have been added in the revised manuscript. Most of your suggested references have also been integrated into the revised manuscript. Thanks very much.

[revised manuscript text omitted]

**37 Keywords:**

| 38 | Drought Africa Precipitation Potential evaporation drought management disaster risk reduction |                                                                        |  |  |  |
|----|-----------------------------------------------------------------------------------------------|------------------------------------------------------------------------|--|--|--|
| 50 |                                                                                               | Jian mpim 1/13/2020 5:44 PM
| 39 |                                                                                               | (Derecear Brought Index, ringh resonation,                             |  |  |  |
| 40 |                                                                                               |                                                                        |  |  |  |
| 41 |                                                                                               |                                                                        |  |  |  |
| 42 |                                                                                               |                                                                        |  |  |  |
| 43 |                                                                                               |                                                                        |  |  |  |
| 44 |                                                                                               |                                                                        |  |  |  |
| 45 |                                                                                               |                                                                        |  |  |  |
| 46 |                                                                                               |                                                                        |  |  |  |
| 47 |                                                                                               |                                                                        |  |  |  |
| 48 |                                                                                               |                                                                        |  |  |  |
| 49 |                                                                                               |                                                                        |  |  |  |
| 50 |                                                                                               |                                                                        |  |  |  |
| 51 |                                                                                               |                                                                        |  |  |  |
| 52 |                                                                                               |                                                                        |  |  |  |
| 53 |                                                                                               |                                                                        |  |  |  |
| 54 |                                                                                               |                                                                        |  |  |  |
| 55 |                                                                                               |                                                                        |  |  |  |
| 56 |                                                                                               |                                                                        |  |  |  |
| 57 |                                                                                               |                                                                        |  |  |  |
| 58 |                                                                                               |                                                                        |  |  |  |
| 59 |                                                                                               |                                                                        |  |  |  |
| 60 |                                                                                               |                                                                        |  |  |  |
|    |                                                                                               |                                                                        |  |  |  |
|    | 2                                                                                             |                                                                        |  |  |  |

**62 1 Introduction**

Drought is a complex phenomenon that affects natural environments and socioeconomic systems in the 63 world (von Hardenberg et al., 2001; Vicente-Serrano, 2007; Van Loon, 2015; Wilhite and Pulwarty, 2017). 64 Impacts include crop failure, food shortage, famine, epidemics and even mass migration (Wilhite et al., 65 2007; Ding et al., 2011; Zhou et al., 2018). In recent years, severe events have occurred across the world, 66 such as the 2003 central Europe drought (García-Herrera et al., 2010), the 2010 Russian drought (Spinoni et 67 al., 2015), the 2011 Horn of Africa drought (Nicholson, 2014), the southeast Australian's Millennium 68 drought (van Dijk et al., 2013; Peng et al., 2019d), the 2013/2014 California drought (Swain et al., 2014), the 69 2014 North China drought (Wang and He, 2015) and the 2015–2017 Southern Africa drought (Baudoin et 70 al., 2017; Muller, 2018). Widespread negative effects of these droughts on natural and socioeconomic 71 systems have been reported afterwards (Wegren, 2011; Arpe et al., 2012; Griffin and Anchukaitis, 2014; 72 Mann and Gleick, 2015; Dadson et al., 2019; Marvel et al., 2019). Thus, there is a clear need to improve our 73 knowledge about the spatial and temporal variability of drought, which provides a basis for quantifying 74 drought impacts and the exposure of society, the economy and the environment over different areas and 75 time-scales (Pozzi et al., 2013; AghaKouchak et al., 2015). 76

Generally, drought is defined as a temporal anomaly characterized by a deficit of water compared with long-77 term conditions (Mishra and Singh, 2010; Van Loon, 2015). Droughts can typically be grouped into five 78 types: meteorological (precipitation deficiency), agricultural (soil moisture deficiency), hydrological (runoff 79 80 and/or groundwater deficiency), socioeconomic (social response to water supply and demand) and environmental or ecologic (Keyantash and Dracup, 2002; AghaKouchak et al., 2015; Crausbay et al., 2017). 81 These different drought categories involve different event characteristics in terms of timing, intensity, 82 duration, and spatial extent, making it very difficult to characterize droughts quantitatively (Panu and 83 Sharma, 2002; Lloyd-Hughes, 2014; Vicente-Serrano, 2016). For this reason numerous drought indices have 84 85 been proposed for precise applications, and reviews of the available indices have been provided by previous studies such as Heim Jr (2002), Keyantash and Dracup (2002), and Mukherjee et al. (2018). Van Loon 86

Jian mpim 1/14/2020 10:44 AM

Jian mpim 1/14/2020 11:11 AM Formatted: Font color: Red

Jian mpim 1/14/2020 10:56 AM Formatted: Font color: Red

Jian mpim 1/13/2020 6:03 PM Formatted: Font color: Red

Jian mpim 1/14/2020 10:44 AM Formatted: Font color: Red

Jian mpim 1/14/2020 10:44 AN Formatted: Font color: Red

(2015) noted that there is no best drought index for all types of droughts, because every index is designed for 87 a specific drought type, thus multiple indices are required to capture the multifaceted nature of drought. 88 Nevertheless, the Standardized Precipitation Index (SPI) is recommended by the World Meteorological 89 Organization (WMO) for drought monitoring, which is calculated based solely on long-term precipitation 90 data over different time spans (McKee et al., 1993). The advantages of SPI are its relative simplicity and its 91 ability to characterize different types of droughts given the different times of response of different usable 92 water sources to precipitation deficits (Kumar et al., 2016; Zhao et al., 2017). However, information on 93 94 precipitation is inadequate to characterize drought; in most definitions, drought conditions also depend on the demand of water vapor from the atmosphere. More recently, Vicente-Serrano et al. (2010) proposed an 95 alternative drought index for SPI, which is called Standardized Precipitation Evapotranspiration Index 96 (SPEI). Compared to SPI, it considers not only the precipitation supply, but also the atmospheric evaporative 97 demand (Beguería et al., 2010; Vicente-Serrano et al., 2012b). This makes the index more informative of the 98 actual drought effects over various natural systems and socioeconomic sectors (Vicente-Serrano et al., 2012b; 99 Bachmair et al., 2016; Kumar et al., 2016; Sun et al., 2016c; Bachmair et al., 2018; Peña-Gallardo et al., 100

101 2018a; Peña-Gallardo et al., 2018b; Sun et al., 2018).

102 For the calculation of SPEI, high-quality and long-term observations of precipitation and atmospheric evaporative demand are necessary. These observations may either come from ground-based station data or 103 104 gridded data such as satellite and reanalysis datasets. For example, the SPEIbase (Beguería et al., 2010) and the Global Precipitation Climatology Centre Drought Index (GPCC-DI) (Ziese et al., 2014) both provide 105 106 SPEI datasets at global scale. The SPEIbase provides gridded SPEI with a 50-km spatial resolution, and is calculated from Climatic Research Unit (CRU) Time-Series (TS) datasets, which are produced based on 107 measurements from more than 4000 ground-based weather stations over the world (Harris et al., 2014). The 108 109 SPEI dataset provided by GPCC-DI has spatial resolution of 1°, and was generated from GPCC precipitation 110 (Becker et al., 2013; Schneider et al., 2016) and National Oceanic and Atmospheric Administration 111 (NOAA)'s Climate Prediction Center (CPC) temperature dataset (Fan and Van den Dool, 2008). Both of Jian mpim 1/14/2020 10:49 AM Formatted: Font color: Red

Jian mpim 1/13/2020 5:50 PM **Deleted:** not enough

Jian mpim 1/14/2020 10:44 AM Formatted: Font color: Red

[revised manuscript text omitted]